# Molecular and clinicopathological implications of *PRAME* expression in adult glioma

**Minh-Khang Le**[1], **Huy Gia Vuong**[2], **Ian F. Dunn**[3], **Tetsuo Kondo**[1]*

**1** Department of Pathology, University of Yamanashi, Chuo City, Yamanashi Prefecture, Japan,
**2** Department of Pathology, University of Iowa Hospitals and Clinics, Iowa City, IA, United States of America,
**3** Department of Neurosurgery, Oklahoma University Health Sciences Center, Oklahoma City, OK, United States of America

* ktetsuo@yamanashi.ac.jp

**Data Availability Statement:** The mRNA, methylation data underlying the results presented in the study are available from GDC (https://portal.gdc.cancer.gov/). The clinical, WES, and WGS data underlying the results presented in the study are

## Abstract

### Background

PRAME (PReferentially expressed Antigen in MElanoma) is a biomarker studied in various human cancers. Little is known about the biological implications of PRAME in glioma. We aimed to perform a comprehensive analysis to explore *PRAME* gene expression and its biological and clinicopathological significance in gliomas.

### Methods and materials

We accessed the human cancer atlas (TCGA) database to collect glioma patients (n = 668) with primary tumors and gene expression data. Single nucleotide variants, copy number variation, DNA methylation data, and other clinicopathological factors were also extracted for the analysis.

### Results

Overall, 170, 484, and 14 tumors showed no expression, low expression (FPKM≤1), and overexpression (FPKM>1) of the *PRAME* gene, respectively. The principal component analysis and pathway analyses showed that *PRAME*-positive gliomas (n = 498), which consisted of tumors with *PRAME* low expression and overexpression, expressed different oncogenic profiles, possessing higher activity of Hedgehog, P3IK-AKT-mTOR, and Wnt/β-catenin pathways (p<0.001). DNA methylation analysis also illustrated that *PRAME*-positive tumors were distributed more densely within a grade 4-related cluster (p<0.001). *PRAME* positivity was an independent prognostic factor for poor outcomes in a multivariate cox analysis adjusted for clinical characteristics and genetic events. Kaplan-Meier analysis stratified by revised classification showed that *PRAME* positivity was solely associated with *IDH*-wild-type glioblastoma, grade 4. Finally, *PRAME*-overexpressing cases (n = 14) had the worst clinical outcome compared to the *PRAME*-negative and *PRAME*-low cohorts (adjusted p<0.001) in pairwise comparisons.

available from cBioPortal (https://www.cbioportal.org/).

**Funding:** The author(s) received no specific funding for this work.

**Competing interests:** The authors have declared that no competing interests exist.

**Abbreviations:** PRAME, Preferentially expressed antigen in melanoma; TCGA, The human cancer atlas; FPKM, fragments per kilobase of exon per million mapped fragments; IDH, isocitrate dehydrogenase; PLA, pigmented lesion assay; IHC, immunohistochemistry; WHO, World Health Organization; GSEA, gene set enrichment analysis; ssGSEA, single-sample gene set enrichment analysis; GSVA, gene set variation analysis.

## Conclusion

*PRAME* expression statuses may dictate different biological and clinicopathological profiles in *IDH*-wildtype glioblastoma.

## Introduction

PRAME (PReferentially expressed Antigen in MElanoma) is a cancer-testis antigen that is expressed by melanoma cells and was isolated by autologous T cells in a melanoma patient [1]. PRAME expression is used to support the diagnosis of melanoma over nevus in combination with histopathological features and other findings. The expression of the *PRAME* gene and PRAME protein can be practically evaluated by pigmented lesion assay (PLA) and immuno-histochemistry (IHC), respectively [2, 3]. Although PRAME has its primary application in the diagnosis of melanoma, PRAME was also found to be expressed by various epithelial and non-epithelial cancers [4], including uterine carcinoma, uterine carcinosarcoma, ovarian carcinoma, adenoid cystic carcinoma, seminoma, thymic carcinoma, basal cell carcinoma, synovial sarcoma, myxoid liposarcoma, and neuroblastoma. The biological and clinicopathological implications of PRAME expression have been unknown in adult gliomas.

Adult gliomas are a heterogeneous and common group of brain cancers with unclear cell-of-origin [5]. The biological profile of gliomas has been studied with respect to histology, epigenetic, genetic characteristics, cell-of-origin, and tumor microenvironment [6, 7]. There are highly diverse oncogenic mechanisms contributing to gliomagenesis and tumor progression, including Wnt/β-catenin [8], PI3K/Akt/mTOR [9], TGF-β [10], and mesenchymal transition, among many others [11]. Important genetic abnormalities affecting the prognosis of glioma patients consist of *IDH1/2* mutations, *CDKN2A/B* homozygous deletion, *EGFR* amplification, *TP53* mutations, *ATRX* mutations, *TERT* promoter mutations, and 7 gain 10 loss chromosomal abnormalities [12]. The recent World Health Organization (WHO) classification of Tumours of the Central Nervous System (CNS) emphasizes that glioma can be divided by *IDH* mutation and 1p/19q codeletion status. *IDH*-wildtype astrocytoma has more advanced clinico-pathological progression and tumor with grade 4 is referred to as the "glioblastoma" category. Glioblastoma is diagnosed by the absence of *IDH* mutation and one of the high-grade features, including high-grade morphology, *TERT* promoter mutation, 7 gain/10 loss chromosomal abnormality, or *EGFR* amplification. However, the prognostic factors of glioma are still under investigation.

In the present study, we first examined how *PRAME* gene expression was related to biological profiles. Secondly, we investigated whether *PRAME* expression patterns were related to the *DNA* methylation landscape. Finally, clinicopathological characteristics and survivorship were compared between *PRAME*-low and *PRAME*-high glioma patients.

## Materials and methods

### Data processing

The Human Cancer Atlas (TCGA) database consists of many datasets. We extracted cases from the TCGA-GBM and TCGA-LGG projects. Only cases with available gene expression profiles (GEP) and primary tumors (no recurrent or metastatic tumors) were included in the study. To adapt to the new WHO classification, cases with grades 2/3 in the previous studies [13, 14] were reclassified into grade 4 as follows: (1) the presence of both *IDH1/2* mutation and

*CDKN2A/B* homozygous deletion, or (2) the absence of *IDH1/2* mutations and the presence of at least one of the following abnormalities: *TERT* promoter mutation, *EGFR* amplification, and 7 gain 10 loss chromosomal abnormality. Other histopathological grading features such as microvascular proliferation and pseudopalisading necrosis were assumed to be included in previously evaluated grade IV gliomas of the original studies [13, 14]. Tumors with the absence of *IDH1/2* mutations and no other high-grade morphological and genetic features mentioned above were reclassified as Astrocytoma, Not Otherwise Specified (NOS). Tumors with the presence of both *IDH1/2* mutations and 1p/19q codeletion were categorized as Oligodendro-glioma. Mixed glioma was re-distributed into new categories based on *IDH1/2* mutation and 1p/19q codeletion status. This reclassification was published in our previous paper [15]. The difference in data processing between this study and our previous one was that we used cbio-portal for cancer genomics (https://www.cbioportal.org) datasets that are related to TCGA-GBM and TCGA-LGG projects, including (1) Brain Lower Grade Glioma (TCGA, Firehose Legacy), (2) Glioblastoma Multiforme (TCGA, Firehose Legacy), and (3) Merged Cohort of LGG and GBM (TCGA, Cell 2016). This difference led to a slight inconsistency in the total number of glioma patients.

### Gene set variation analysis (GSVA)

GSVA is a type of single-sample gene set enrichment analysis (ssGSEA) [16], which is a variant of conventional GSEA [17]. The rank of genes was based on an expression-level statistic, which is a Gaussian or Poisson kernel estimation of the cumulative density function of each gene across the samples. Each value of kernel estimation was calculated, given each gene expression of each sample. As a result, the enrichment score (ES) of a gene set can be calculated for each sample. Therefore, we can investigate the activity of signaling pathways, using GSVA. In this study, we employed the gene sets of the hallmark pathways ("H" category) in the MSigDB database (https://www.gsea-msigdb.org) and excluded cancer-irrelevant pathways.

### Data analysis

Continuous and categorical variables were described by median (range) and the number of cases (percentage), respectively. Chi-square tests and Wilcoxon's test were performed to compare categorical and continuous variables by default. Pathway activity was calculated by GSVA. For survival analysis, we performed Kaplan-Meier analysis and univariate and multi-variate Cox analysis. Heatmap was created, using the Biokit package, run on Python 3.9. Other data analyses were conducted, using R version 4.2.1 (The R Foundation, Austria).

## Results

### Investigating *PRAME* expression and pathway activities

A total of 668 primary tumors in 668 glioma patients, were included in our study. We explored the normalized read count value of these 668 gliomas and found that the *PRAME* gene was not expressed in 170 gliomas (*PRAME*-negative; FPKM = 0). **Fig 1A** shows the normalized read counts of the *PRAME* expression. Therefore, the entire cohort was divided into two groups, *PRAME*-negative and *PRAME*-positive, based on the *PRAME* expression status. Mean *PRAME* gene expression in the *PRAME*-positive cohort was 0.98 FPKM, which was relatively low. Most cases (n = 654) had *PRAME* gene expression < 1 FPKM, while only 14 cases had high *PRAME* expression (**Fig 1A, inset**). Next, we conducted principal component analysis (PCA) over 730 genes within the nCounter PanCancer Pathways panel, published by Nanostring Technology (https://nanostring.com/, excluding 40 internal reference genes). This panel

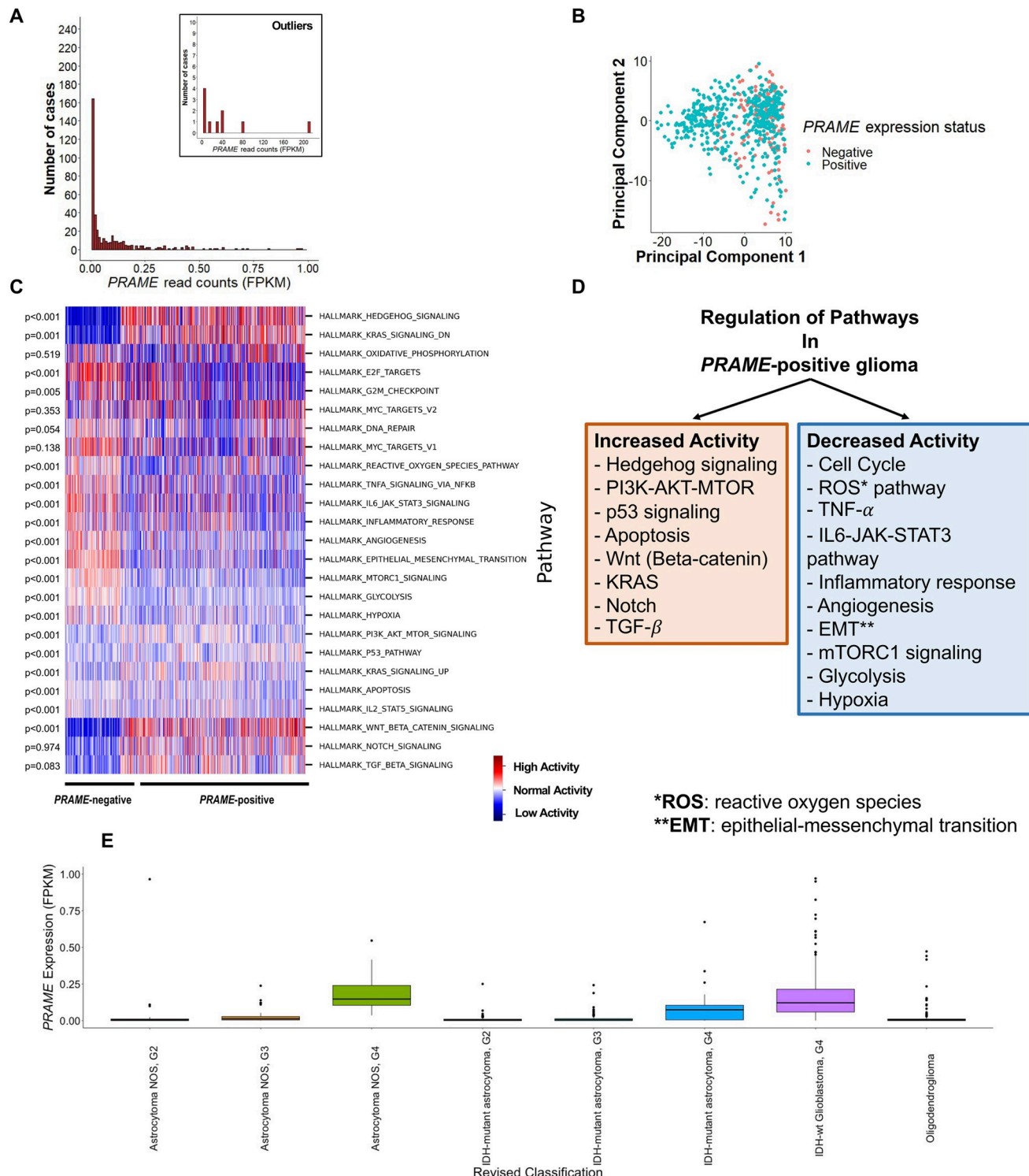

**Fig 1.** (A) Histogram shows the distribution of the *PRAME* expression read counts (FPKM). The inset only shows the distribution of tumors with FPKM>1. (B) Principal component analysis (PCA) plot created by PC1 and PC2 of the nCounter Nanostring PanCancer Pathways gene panel. (C) The activity heatmap of cancer-related pathways illustrates the activity of each pathway in each sample. The p-values in the left column are calculated by the two-sample independent t-tests to compare the enrichment score (ES) of each pathway between *PRAME*-negative and *PRAME*-positive tumors. (D) The summary plot of upregulated and downregulated pathways in *PRAME*-positive gliomas. (E) The boxplot compares *PRAME* expression between different revised categories of the new WHO classification.

included genes within 13 cancer-associated canonical pathways, which supports the understanding of basic cancer biology. **S1 Table** shows the symbol and ID of the genes within the panel. A dimension reduction plot (**Fig 1B**) was created, using PC1 and PC2 components, which explained the highest variance in the data. We found that *PRAME*-negative tumors clustered together with some *PRAME*-positive counterparts. However, there was a higher percentage of *PRAME*-positive gliomas outside of this cluster. Therefore, there were differences in the distributions of *PRAME*-negative and *PRAME*-positive gliomas in the 730-gene cancer-related space. Finally, we created a heatmap to show the pathway-level comparisons between the two groups, using two-sample independent t-tests (**Fig 1C**). The *PRAME*-positive tumors expressed higher activity of Hedgehog ($p < 0.001$), P3IK-AKT-mTOR ($p < 0.001$), P53 ($p < 0.001$), apoptosis ($p < 0.001$), IL2-STAT5 ($p < 0.001$), and Wnt/β-catenin ($p < 0.001$) signaling pathways while these tumors reduced other biological signals, including E2F targets ($p < 0.001$), G2M mitotic checkpoint ($p = 0.005$), reactive oxidative oxygen species ($p < 0.001$), TNF-α ($p < 0.001$), IL6-JAK-STAT6 ($p < 0.001$), inflammatory response ($p < 0.001$), angiogenesis ($p < 0.001$), epithelial-mesenchymal transition ($p < 0.001$), mTORC1 ($p < 0.001$), glycolysis ($p < 0.001$), and hypoxia ($p < 0.001$). **Fig 1D** summarizes the results of pathway analysis. In addition, **Fig 1E** illustrates the *PRAME* expression in each revised category. In general, gliomas grade 4 had higher *PRAME* expression compared to other categories.

### *PRAME* positivity was densely distributed within a distinct DNA methylation cluster

In the study cohort, there were a total of 476 tumors with available data about 450K DNA methylation. We performed t-SNE dimension reduction to explore the differences in the distribution of *PRAME*-negative and *PRAME*-positive glioma in DNA methylation hyperspace. This DNA methylation space can be interpreted as the reduced representation of the CpG methylation landscape. We also included the Glioma CpG Island Methylator Phenotype (G-CIMP), which was published in a previous paper [33941250]. In this study, 476 tumors in DNA methylation space can be relatively divided into two unsupervised tSNE clusters, small (right, lower corner) and large clusters (left and upper part) (**Fig 2A–2D**). The small cluster densely consisted of *PRAME*-low and *PRAME*-overexpressing samples while the larger cluster had a significant portion of *PRAME*-negative samples (**Fig 2A**). The revised subtype (**Fig 2B**), CIMP clusters (**Fig 2C**), and *IDH* status (**Fig 2D**) were strongly associated with these 2 clusters. **Fig 2E** shows a heatmap of distribution of *PRAME* expression status within CIMP clusters. There were significant difference in the distribution of *PRAME* expression status (chi-square test, $p < 0.001$). This discrimination can be seen in LGm6-GBM (6/12 vs. 7/484 vs. 0/170), classic-like (1/12 vs. 63/484 vs. 4/170), and mesenchymal-like (2/12 vs. 87/484 vs. 9/170).

### *PRAME* positivity was associated with *IDH*-wildtype glioblastoma and adverse outcomes

**Table 1** summarizes the clinicopathological characteristics of *PRAME*-negative and *PRAME*-positive gliomas. Clinically, patients with *PRAME*-positive gliomas were older ($p < 0.001$). There were no differences in gender ($p = 0.419$) and race ($p = 0.382$). Comparisons of revised classification between *PRAME*-negative and *PRAME*-positive cohorts showed that *IDH*-mutant astrocytoma, grade 2 (26.5% vs. 13.1%) and oligodendroglioma (35.9% vs. 21.7%) dominated *PRAME*-negative glioma while the incidence of *IDH*-wildtype glioblastoma, grade 4 (32.9% vs. 4.1%) was much higher in *PRAME*-positive glioma. These differences were significant ($p < 0.001$). Regarding genetic abnormalities, *PRAME*-positive tumors more frequently acquired *EGFR* amplification (20.4% vs. 4.7%; $p < 0.001$), *CDKN2A/B* homozygous deletion

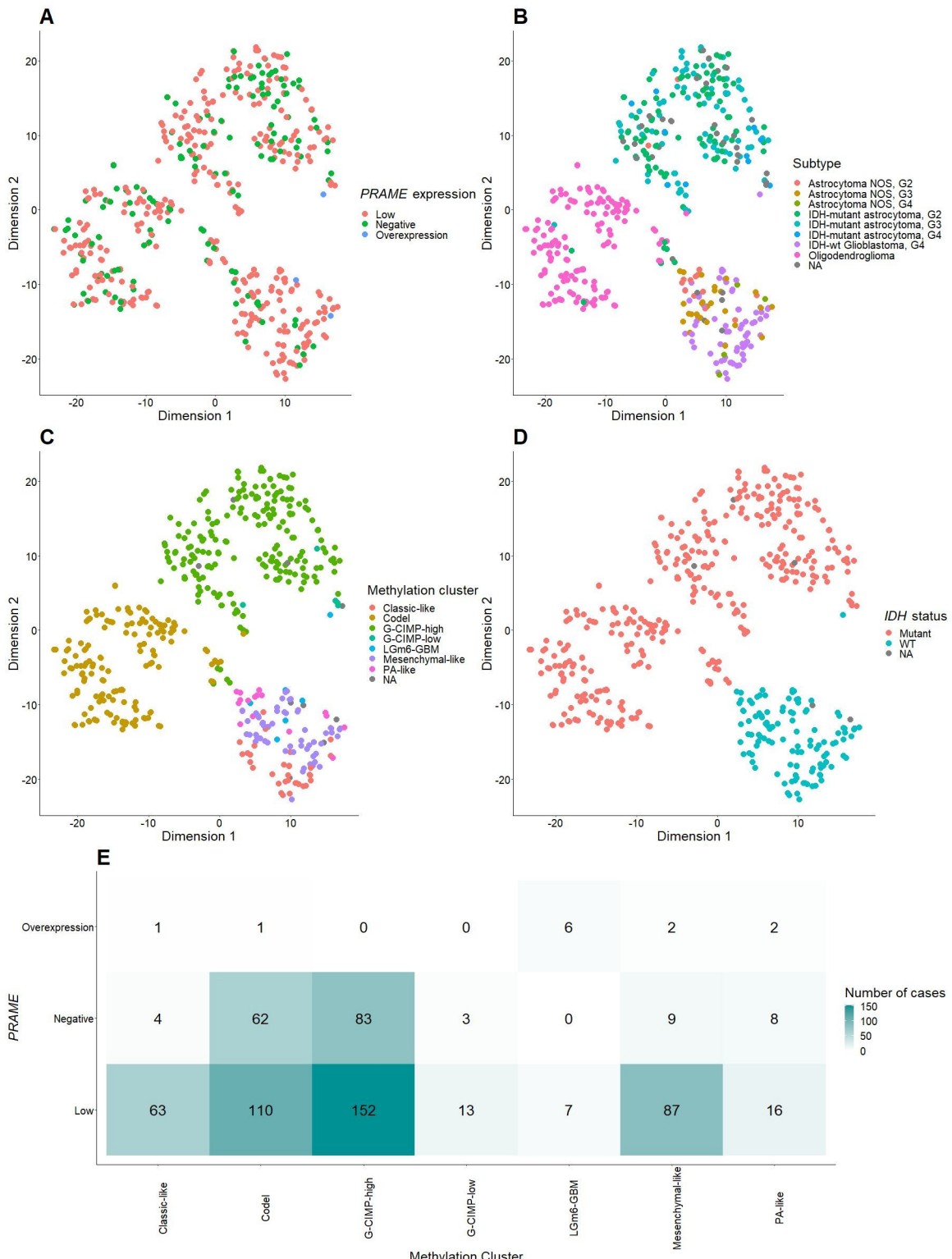

**Fig 2.** t-SNE dimension reduction plots show the distribution of DNA methylation landscapes of 474 tumors, characterized by *PRAME* expression status (A), re-classified WHO grades (B), and *IDH* mutation status (C).

**Table 1. Comparison of clinicopathological characteristics of *PRAME*-negative and *PRAME*-positive cohorts.**

| Variable | *PRAME*-negative (n = 170) | *PRAME*-positive (n = 498) | p-value |
|---|---|---|---|
| **Age** | 39 (17–74) | 4 (14–89) | <0.001 |
| **Gender** | | | 0.419 |
| Female | 68 (44.7%) | 185 (40.6%) | |
| Male | 84 (55.3%) | 271 (59.4%) | |
| **Race** | | | 0.382 |
| White | 155 (91.2%) | 455 (91.4%) | |
| Asian | 2 (1.2%) | 11 (2.2%) | |
| Black | 7 (4.1%) | 24 (4.8%) | |
| Not reported | 6 (3.5%) | 8 (1.6%) | |
| **Revised classification** | | | <0.001 |
| Astrocytoma NOS, grade 2 | 8 (4.7%) | 12 (2.4%) | |
| Astrocytoma NOS, grade 3 | 7 (4.1%) | 28 (5.6%) | |
| Astrocytoma NOS, grade 4 | 0 (0.0%) | 8 (1.6%) | |
| *IDH*-mutant astrocytoma, grade 2 | 45 (26.5%) | 65 (13.1%) | |
| *IDH*-mutant astrocytoma, grade 3 | 26 (15.3%) | 69 (13.9%) | |
| *IDH*-mutant astrocytoma, grade 4 | 3 (1.8%) | 18 (3.6%) | |
| *IDH*-wildtype glioblastoma, grade 4 | 7 (4.1%) | 164 (32.9%) | |
| Oligodendroglioma | 61 (35.9%) | 108 (21.7%) | |
| Unknown | 13 (7.6%) | 26 (5.2%) | |
| ***IDH1/2* mutation** | 148/169 (87.6%) | 279/492 (56.7%) | <0.001 |
| ***TP53* mutation** | 76/170 (44.7%) | 207/497 (41.6%) | 0.545 |
| ***ATRX* mutation** | 68/170 (40.0%) | 136/497 (27.4%) | 0.005 |
| ***TERT* promoter mutation** | 37/91 (40.7%) | 117/227 (51.5%) | 0.103 |
| ***EGFR* amplification** | 8/169 (4.7%) | 100/490 (20.4%) | <0.001 |
| ***CDKN2A/B* homozygous deletion** | 11/169 (6.0%) | 125/490 (25.5%) | <0.001 |
| **7 gain 10 loss** | 12/169 (7.1%) | 139/490 (28.4%) | <0.001 |
| **Vital status** | | | <0.001 |
| Alive | 129 (84.9%) | 298 (65.4%) | |
| Dead | 23 (15.1%) | 158 (34.6%) | |
| **Overall survival time (months)** | 15.2 (0.0–134.0) | 11.5 (0.0–211.0) | 0.087 |

NOS: Not otherwise specified.

(25.5% vs. 6.0%; p<0.001), and 7 gain 10 loss chromosomal aberrations (28.4% vs. 7.1%; p<0.001) than *PRAME*-negative tumors. Conversely, *IDH1/2* mutations (87.6% vs. 56.7%; p<0.001) and *ATRX* mutations (40.0% vs. 27.4%; p = 0.005) were significantly more common in *PRAME*-negative cases.

In survival analysis, *PRAME* was a general marker of prognosis in the entire studied cohort (p<0.001, **Fig 3A**). Stratified by the new WHO classification, there were no significant results in *IDH*-mutant astrocytoma grade 2 (p = 0.891, **Fig 3B**), grade 3 (p = 0.502, **Fig 3C**), and grade 4 (p = 0.160, **Fig 3D**). However, *PRAME* positivity was of prognostic significance in *IDH*-wild-type glioblastoma grade 4 (p = 0.018, **Fig 3E**). There was no obvious survival difference between *PRAME*-positive and *PRAME*-negative oligodendrogliomas. We also compared the survival outcomes of *IDH*-mutant/1p19q codeletion and *IDH*-mutant/non-1p19q codeletion (**S1 Fig**) but *PRAME* positivity was not related to the prognosis. **Table 2** shows multivariate analyses adjusted for clinical characteristics, whole genome sequencing (WGS) (*IDH1/2* mutation, *ATRX* mutation, and *TP53* mutation), whole exome sequencing (WES) (*TERT* promoter

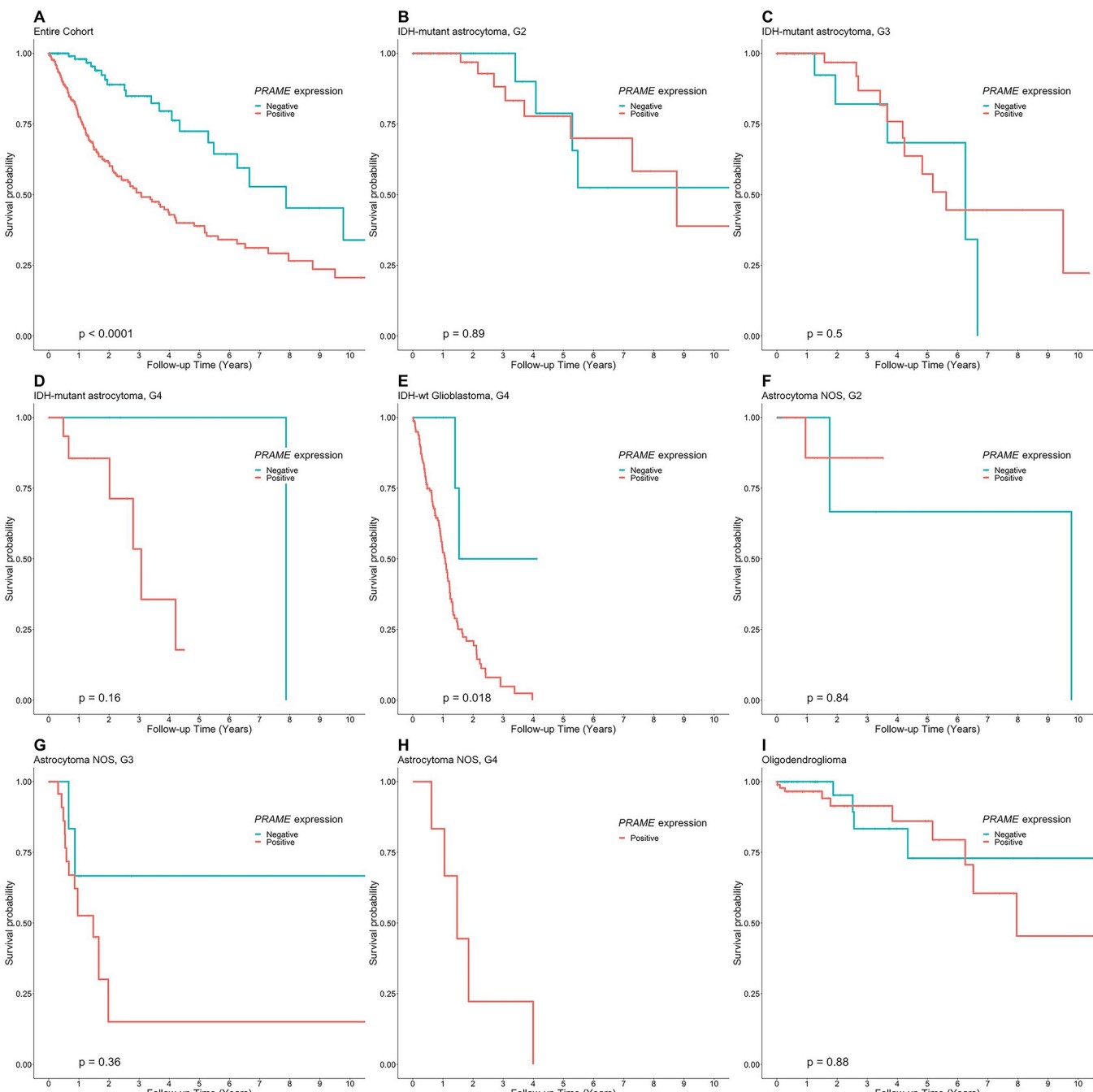

**Fig 3.** Kaplan-Meier curves illustrate the different survival patterns of *PRAME*-negative and *PRAME*-positive tumors in the entire cohort (A), IDH-mutant astrocytoma, grade 2 (B), IDH-mutant astrocytoma, grade 3 (C), and IDH-mutant astrocytoma, grade 4 (D), *IDH*-wildtype glioblastoma (E), astrocytoma, NOS, grade 2 (F), astrocytoma, NOS, grade 3 (G), astrocytoma, NOS, grade 4 (H), and oligodendroglioma (I).

mutaton), and copy number variation information (*CDKN2A/B* homozygous deletion, *EGFR* amplification, and 7 gain/10 loss). The prognostic effect of *PRAME* positivity was significant and independent to clinical characteristics (HR = 2.73; 95%CI = 1.66–4.15; p<0.001), WGS (HR = 2.03; 95%CI = 1.28–3.22; p = 0.003), and CNV (HR = 2.11; 95%CI = 1.35–3.29; p = 0.001) features.

**Table 2. Multivariate Cox survival analyses with overall survival and vital status as the outcome, adjusted for clinical characteristics, WGS, WES, and CNV information.**

| Variable | HR | 95%CI | p-value |
|---|---|---|---|
| **Clinical characteristics** | | | |
| *PRAME* positive | 2.73 | 1.66–4.50 | <0.001 |
| Age (years) | 1.08 | 1.06–1.09 | <0.001 |
| Men | 1.05 | 0.76–1.45 | 0.761 |
| Race | | | |
| Asian | 1 | | |
| Black | 0.79 | 0.16–3.93 | 0.769 |
| White | 0.73 | 0.18–2.98 | 0.662 |
| **WGS-available data** | | | |
| *PRAME* positive | 2.03 | 1.28–3.22 | 0.003 |
| *IDH1/2* mutation | 0.07 | 0.05–0.12 | <0.001 |
| *ATRX* mutation | 1.56 | 0.85–2.87 | 0.151 |
| *TP53* mutation | 1.09 | 0.75–1.59 | 0.657 |
| **WES-available data** | | | |
| PRAME positive | 1.61 | 0.94–2.77 | 0.083 |
| *TERT* promoter mutation | 2.02 | 1.28–3.20 | 0.003 |
| **CNV-available data** | | | |
| PRAME positive | 2.11 | 1.35–3.29 | 0.001 |
| *EGFR* amplification | 1.36 | 0.90–2.06 | 0.143 |
| *CDKN2A/B* homozygous deletion | 2.55 | 1.70–3.81 | <0.001 |
| 7 gain/10 loss | 3.61 | 2.35–5.55 | <0.001 |

HR: hazard ratio; 95%CI: 95% confidence interval; WGS: whole genome sequencing; WES: whole exome sequencing; CNV: copy number variation.

## *PRAME* overexpression was associated with a worse prognosis than *PRAME* positivity

First, we defined *PRAME* overexpression when FPKM > 1, which was observed in only 14 gliomas within the entire studied cohort. **Table 3** shows the clinicopathological characteristics of these 14 patients. Notably, only 1 case (7.1%) of this cohort was *IDH*-mutant. Next, we performed KM analysis to compare the overall survival of *PRAME*-negative (n = 170), *PRAME*-low (n = 481), and *PRAME*-overexpressing (n = 14) cases (*PRAME*-low and *PRAME*-overexpressing comprise *PRAME*-positive cohort) (**Fig 4**). To avoid false-positive results, pairwise log-rank test comparisons were conducted, using Benjamini-Hochberg correction to calculate the adjusted p-values. Even with a small sample (n = 14), *PRAME*-overexpressing glioma showed a significantly worse outcome than *PRAME*-low (adjusted p<0.001), and *PRAME*-negative (adjusted p<0.001). 2-year overall survival rates of *PRAME*-negative, *PRAME*-low, and *PRAME*-overexpressing cohorts were 92.8% (95%CI = 88.1% - 97.8%), 64.0% (95% CI = 59.2% - 69.1%), and 13.8% (95%CI = 2.6% - 73.3%), respectively.

## Analysis of the association between tumor microenvironment (TME) and *PRAME* expression

Given that *PRAME* is associated with cytotoxic T-cell activation and killing in glioblastoma [18], we compared the immunologic cell population between *PRAME*-negative and *PRAME*-positive gliomas. GSVA of cell type-specific gene sets [19] was performed to calculate the

**Table 3. Clinicopathological data of 14 patients with *PRAME*-overexpressing gliomas.**

| Patient ID | Age (yo) | Gender | Race | Revised classification | PRAME expression (FPKM) | Mutations | | | | | | | OS time (months) | Vital Status |
|---|---|---|---|---|---|---|---|---|---|---|---|---|---|---|
| | | | | | | IDH1/2 | TP53 | ATRX | TERT | CHD | EGFRamp | 7+/10- | | |
| TCGA-02-0047 | 79 | M | White | IDH-wt Glioblastoma, G4 | 37.9 | No | No | No | No | Yes | No | No | 14.9 | Dead |
| TCGA-06-0168 | 60 | F | White | IDH-wt Glioblastoma, G4 | 1.3 | No | No | No | No | No | Yes | No | 19.9 | Dead |
| TCGA-06-0646 | 61 | M | White | IDH-wt Glioblastoma, G4 | 2.6 | No | No | No | No | Yes | Yes | No | 5.8 | Dead |
| TCGA-06-2569 | 24 | F | Black | IDH-wt Glioblastoma, G4 | 211.7 | No | Yes | No | No | No | Yes | No | 0.4 | Alive |
| TCGA-06-5411 | 52 | M | White | IDH-wt Glioblastoma, G4 | 1.1 | No | No | No | No | Yes | Yes | No | 8.5 | Dead |
| TCGA-12-0821 | 63 | M | White | IDH-wt Glioblastoma, G4 | 41.9 | No | No | No | No | Yes | Yes | No | 10.8 | Dead |
| TCGA-14-0871 | 75 | F | White | IDH-wt Glioblastoma, G4 | 32.2 | No | Yes | No | No | n/a | n/a | n/a | 29.3 | Dead |
| TCGA-26-5133 | 59 | M | White | IDH-wt Glioblastoma, G4 | 1.5 | No | Yes | No | No | No | No | Yes | 15.1 | Alive |
| TCGA-28-5218 | 63 | M | White | Astrocytoma NOS, G3 | 17.3 | No | No | No | No | Yes | Yes | No | 5.2 | Dead |
| TCGA-DH-5140 | 38 | F | White | IDH-wt Glioblastoma, G4 | 6.5 | No | Yes | No | No | No | Yes | No | 20.2 | Dead |
| TCGA-DU-6403 | 60 | F | White | IDH-wt Glioblastoma, G4 | 6.9 | No | No | No | No | No | Yes | Yes | 11.8 | Dead |
| TCGA-E1-A7YD | 58 | M | White | IDH-wt Glioblastoma, G4 | 3.9 | No | Yes | No | No | No | Yes | No | 14.5 | Dead |
| TCGA-FG-5963 | 23 | M | White | IDH-wt Glioblastoma, G4 | 79.5 | No | Yes | Yes | No | Yes | No | No | 25.8 | Dead |
| TCGA-S9-A7IY | 40 | M | White | Oligodendroglioma | 1.3 | Yes | No | No | No | No | No | No | 23.8 | Alive |

CHD: *CDKN2A/B* homozygous deletion; *EGFR*amp: *EGFR* gene amplification; 7+/10-: 7 gain 10 loss chromosomal abnormalities; OS: overall survival.

activity (ES) of 17 cell types, including B cells, T cells, T helper, Th1, Th2, TFH, Th17, Treg, CD8 T cells, T gamma delta, cytotoxic cells, NK cells, dendritic cells, eosinophils, macrophages, mast cells, and neutrophils (**Fig 5**). There were increased activities of T cells (p<0.001), Th2 (p<0.001), Th17 (p<0.001), cytotoxic cells (p = 0.028), macrophages (p<0.001), and neutrophils (p<0.001) in *PRAME*-positive gliomas while there was reduced activities of TFH (p<0.001) and CD8 T cell (p<0.001).

## Discussion

In the present study, we showed that *PRAME* expression status was significantly correlated with biological and clinicopathological characteristics of adult glioma grade 4, *IDH*-wildtype (*IDH*-wildtype glioblastoma). In gene expression analysis, there was a large number of gliomas showing no *PRAME* expression while a few numbers of tumors possessed high levels of *PRAME* expression. The remaining tumors generally showed low *PRAME* expression. Therefore, the included gliomas were divided into *PRAME*-negative and *PRAME*-positive subgroups. We then compare biological profiles and clinicopathological characteristics between *PRAME*-negative and *PRAME*-positive cases. The PCA of the PanCancer Pathways panel showed that different *PRAME* expression status was relatively different in their distributions,

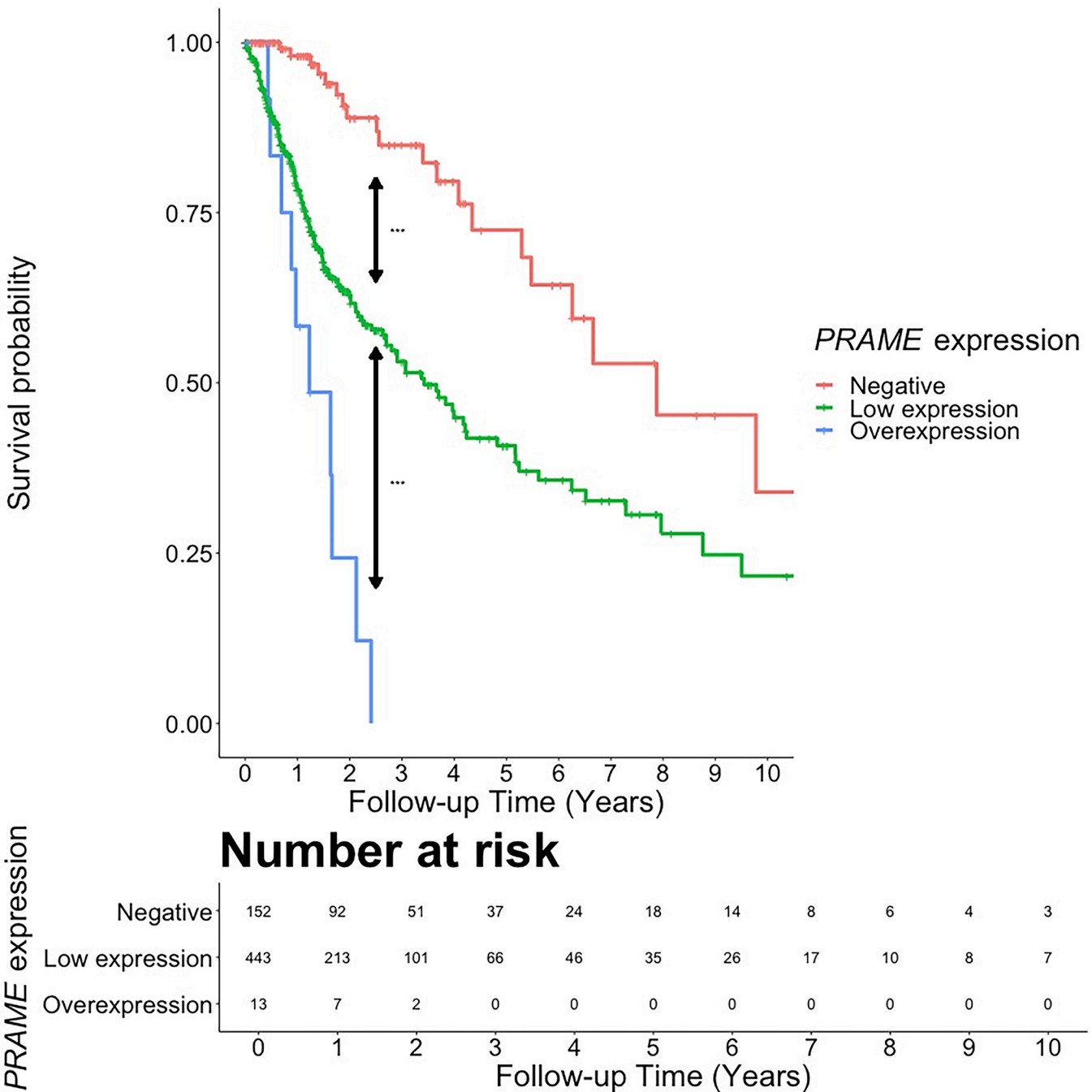

**Fig 4. The Kaplan-Meier curve shows the pairwise comparisons of survivorship between *PRAME*-negative, *PRAME* low-expressing, and *PRAME*-overexpressing tumors.** (\*\*\*), (\*\*), and (\*) indicate adjusted p < 0.001, adjusted p = < 0.01, and adjusted p < 0.05, respectively. The risk table illustrates the number of cases that survived across the timeline in each cohort of *PRAME* expression status.

indicating that *PRAME* positivity may be related to oncogenic mechanisms in adult glioma. In pathway analysis, we illustrated that *PRAME*-positive gliomas possessed higher activity of Hedgehog, P3IK-AKT-mTOR, P53, apoptosis, IL2-STAT5, and Wnt/β-catenin signaling pathways and lower expression of E2F targets, G2M mitotic checkpoint, reactive oxidative oxygen species, TNF-α, IL6-JAK-STAT6, inflammatory response, angiogenesis, epithelial-

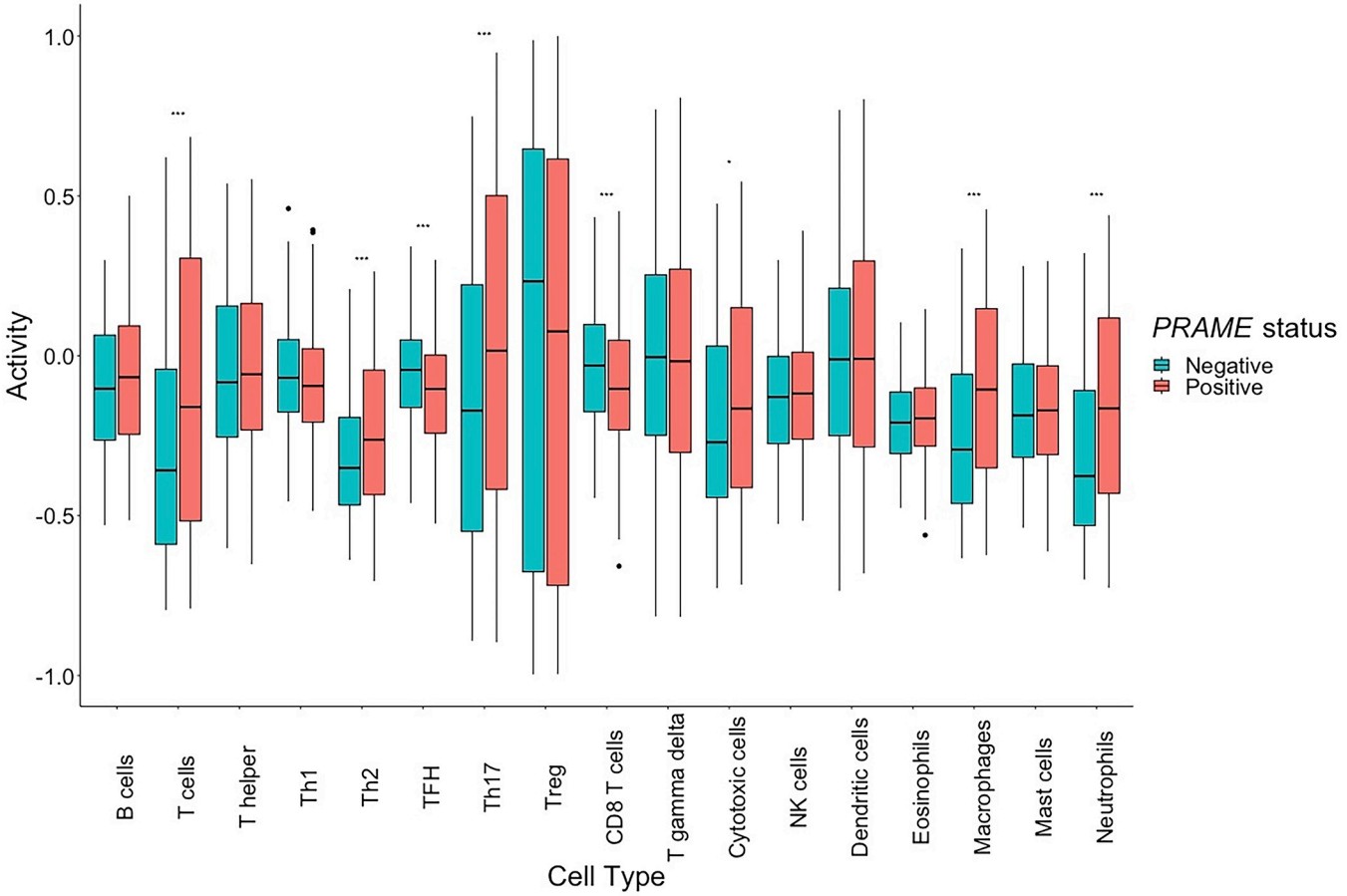

**Fig 5. The boxplots comparing activity of different immunologic cell populations between *PRAME*-negative and *PRAME*-positive gliomas.**

mesenchymal transition, mTORC1, glycolysis, and hypoxia. In DNA methylation analysis, *PRAME*-positive gliomas were distributed more densely in a distinct, grade 4-related cluster, which implied that *PRAME* expression can be an indicator of the CpG methylation landscape. Clinicopathologically, *PRAME* positivity was associated with older age, higher grades, *EGFR* amplification, *CDKN2A/B* homozygous deletion, and 7 gain 10 loss. This association was also related to *IDH*-wildtype glioblastoma in the present study. Finally, *PRAME* expression status was identified as an independent prognostic factor of *IDH*-wildtype glioblastoma.

The significance of *PRAME* expression has been mentioned previously. Wu et al. [20] developed a *PRAME*-containing formula for risk score, which was inferred from the regression model. This score quantified the risk of Karnofsky performance score, but not the prognosis itself. Therefore, the inference of *PRAME* prognostic value is plausible but weak. The other study by Zhang et al. [21] mainly compared the *PRAME* expression between different types of brain tumors, including subtypes of astrocytic and non-astrocytic tumors. However, it can be difficult to conclude that *PRAME* expression is an independent prognostic factor by the current evidence. On the other hand, the goal of our study is to focus on the biological and clinicopathological characteristics of *PRAME*. Multiple analyses were performed to provide more concrete proof of the significance of this gene in glioma.

PRAME has been recently introduced as a prognostic and/or oncogenic biomarker of various cancer types, including melanocytic neoplasms [22], invasive breast carcinoma [23], lung

adenocarcinoma [24], lung squamous cell carcinoma [25], and hematological malignancies [26]. PRAME has also been found to be expressed by various types of neoplasms as mentioned earlier [4]. Regarding CNS tumors, the significance of *PRAME* expression has been investigated in medulloblastoma [27, 28] as a biomarker for immunotherapy. However, little is known about the biological and clinical significance of the PRAME protein and its corresponding gene in glioma. To the best of our knowledge, the intensity and pattern of PRAME and its gene expression in glioma are still under investigation. In the present study, we found that most of these tumors still expressed *PRAME* at a low level and a minority of them, however, showed gene overexpression. Even with low expression, the *PRAME*-expressing glioma still had distinct biological characteristics, which was shown in subsequent analyses. Further studies to validate PRAME protein expression in glioma, using western blot analysis, immuno-histochemistry, immunofluorescence, or other techniques, are needed because it is not clear whether *PRAME* gene expression can be an indicator of its protein status.

The nCounter Nanostring PanCancer Pathways panel was used to evaluate the biological profiles of human cancers in previous studies [29, 30]. Although the experimental pipeline of Nanostring technology was not performed in the present study, the biological value of the genes should be similar in principle. Using this panel, our PCA analysis illustrated that *PRAME* expression status can be a biomarker of glioma biology although further interpretations are not available in such general results. Pathway analysis showed more details in the biological difference among *PRAME* expression statuses. In glioma, the Wnt/β-catenin signal promotes neurogenesis and cell proliferation while the PI3K/AKT/mTOR pathway is associated with growth, metabolism, autophagy, survival, and chemotherapy resistance of glioblastoma [31]. The hedgehog signaling pathway is also required for glioma-initiating cell proliferation and tumorigenesis [32]. These pathways were increased in *PRAME*-positive gliomas. However, various oncogenic processes or signals in *PRAME*-positive gliomas such as E2F targets, G2M mitotic checkpoint, reactive oxidative oxygen species, IL6-JAK-STAT6, angiogenesis, epithelial-mesenchymal transition, and mTORC1 were activated at the lower levels compared to *PRAME*-negative tumors. These results were controversial, suggesting biological heterogeneity in *PRAME*-positive tumors.

Recent studies showed dozens of clinicopathological risk factors with prognostic significance in adult gliomas. Clinically, age, tumor size, and tumor location within CNS are predictive factors of glioma patient outcomes [33, 34]. Pathologically, histological glioma subtypes and WHO grade are also related to glioma prognosis. Regarding genetic abnormalities, *CDKN2A/B* homozygous deletion, *EGFR* amplification, *TP53* mutations, *ATRX* mutations, *TERT* promoter mutations, and 7 gain 10 loss chromosomal abnormalities are associated with poor prognosis while *IDH1/2* mutations are closely related to superior outcomes [12]. Regarding gene expression, a previous study developed a stemness index from the regularized cox model to predict the prognosis of glioma patients [35]. Therefore, it is important to show the prognostic significance of a biomarker by adjusting such confounders in a multivariate analysis. In the present study, we found that *PRAME* positivity was an independent prognostic factor to other clinicopathological factors. Interestingly, we found that *PRAME* gene overexpression, which is more likely visualized by the protein expression detection methods, was related to a subgroup with a significantly worse prognosis than *PRAME*-low gliomas despite its small sample size.

DNA methylation profile was proven to be pathologically associated with CNS tumors. A methylation-based random forest classifier was developed to provide a novel biological fingerprint of CNS tumors in addition to other identifiers such as histopathology and genetic abnormalities [36]. Another study also argued that methylation profiling can be a reliable biomarker for further low-grade glioma subtyping [37]. Therefore, examining whether there is a

relationship between *PRAME* expression status and DNA methylation characteristics can further consolidate *PRAME* value in glioma biology. Our study showed that *PRAME*-positive gliomas were distributed more densely in the IDH-wildtype-related methylation cluster compared to the other cluster. Although this specific distribution of *PRAME*-positive tumors can be attributed to the dense clustering of grade 4 tumors, we believe that *PRAME* positivity and negativity can still be an indicator of DNA methylation profile, regardless of the causal relationships.

*PRAME* can also be associated with glioma TME. A previous study showed that Decitabine can increase *PRAME* expression and, thus, enhance the T-cell-mediated cytotoxicity, which makes *PRAME* an interesting target for immunotherapy [18]. However, TME of cell line can be difficult to interpret because the stromal or microenvironment context of cancer in vivo is different from that of cell line condition. Our study showed that *PRAME* higher expression was related to increased cytotoxic cell, macrophage, and neutrophil activity but it was also associated with many immune modulating cells such as Th2, Th17, and TFH. Therefore, *PRAME* expression is in a complicated relationship with many immunologic cell populations, not only cytotoxic T cells.

However, there were limitations in the present study. First, selection bias was a potential problem because this study used a public database. Second, our findings of *PRAME* positivity in *IDH*-mutant glioma were not significant potentially due to small samples, and, thus, sampling error. Therefore, a larger study of *PRAME* expression on *IDH*-mutant glioma can be helpful to examine the biological and clinicopathological relevance of *PRAME* positivity in these brain tumors. Third, protein expression data was not fully available and, therefore, cannot be analyzed. *PRAME* gene expression may be different from PRAME protein expression, which can be practically evaluated by immunohistochemistry. Hence, immunohistochemical studies are required to validate PRAME prognostic significance at the protein level. Additionally, data about histone modification is not available in TCGA-LGG and TCGA-GBM projects. Therefore, we were not able to analyze the relationship between *PRAME* expression and this epigenetic regulation. Finally, to our knowledge, there was no available information of chemotherapy and radiotherapy resistance in TCGA datasets. Hence, we were not able to investigate the relationship between *PRAME* expression and treatment response.

## Conclusion

Our study illustrated that a proportion of glioma did not express *PRAME* while the majority of glioma expressed *PRAME*, among which a few tumors possessed high *PRAME* expression. *PRAME*-positive tumors had different biological (gene expression, DNA methylation, and pathway) and clinicopathological characteristics, which were related to *IDH*-wildtype glioblastoma. In survival analysis, *PRAME* positivity, especially *PRAME* overexpression, was related to poor prognosis.

## Supporting information

**S1 Fig.** Kaplan-Meier curves compare the survivorship of *PRAME*-positive and *PRAME*-negative tumors in *IDH*-mutant gliomas with (**A**) and without (**B**) 1p/19q co-deletion.
(TIF)

**S1 Table. Nanostring PanCancer pathways panel consists of 770 genes.**
(XLSX)

## Author Contributions

**Conceptualization:** Minh-Khang Le, Huy Gia Vuong, Ian F. Dunn, Tetsuo Kondo.

**Data curation:** Minh-Khang Le.

**Formal analysis:** Minh-Khang Le, Huy Gia Vuong.

**Investigation:** Minh-Khang Le.

**Methodology:** Minh-Khang Le, Huy Gia Vuong.

**Project administration:** Minh-Khang Le, Tetsuo Kondo.

**Resources:** Minh-Khang Le.

**Visualization:** Minh-Khang Le.

**Writing – original draft:** Minh-Khang Le.

**Writing – review & editing:** Huy Gia Vuong, Ian F. Dunn, Tetsuo Kondo.

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
