## [Decision Letter · Decision Letter 0]

26 Jun 2023

PONE-D-23-03723Molecular and clinicopathological implications of PRAME expression in adult gliomaPLOS ONE

Dear Dr. Kondo,

Thank you for submitting your manuscript to PLOS ONE. After careful consideration, we feel that it has merit but does not fully meet PLOS ONE’s publication criteria as it currently stands. Therefore, we invite you to submit a revised version of the manuscript that addresses the points raised during the review process.

Here is my feedback and the required changes for acceptance:

Reviewer 1 has provided positive feedback on your manuscript and suggests accepting it with some minor suggestions for improvement. I agree with their assessment and require you to address these minor suggestions to enhance the clarity and presentation of your work.

Reviewer 2, on the other hand, has raised concerns about the methodology used in your study and suggests major changes to address potential confounding factors. They recommend revising the manuscript by revisiting the previous similar findings to strengthen the validity of your findings. I concur with their assessment and recommend implementing their major suggestions to improve the scientific rigor of your study.

In addition, Reviewer 2 also suggests some minor changes, such as including additional references, figure legends, and text updates to support your findings. While these changes are not mandatory for acceptance, I encourage you to consider them as they would further strengthen the discussions of your study. He further highlighted some language and formatting issues in the manuscript. I agree with their observations, and I recommend thoroughly revising the manuscript for clarity, grammar, and consistent formatting.

Based on the reviewers' feedback and my evaluation, I am inclined to accept your manuscript pending the revisions mentioned above. These changes are necessary to address the concerns raised by Reviewer 2 and improve the overall quality of your work. Please address all the reviewers' comments and provide a point-by-point response in the cover letter when submitting the revised manuscript.

In your point-by-point responses to the reviewer's comments, please state the page number and paragraph of the tracked manuscript where changes have been made as a result. It can be helpful to tabulate your responses with columns labelled (left to right) as follows: Reviewer comments; author response and changes made; page number in revised, tracked paper where the change can be found.

IMPORTANT: Where a reviewer has asked for clarification, it is usually necessary to amend the manuscript as well as answering the question directly in the point-by-point response document - where no changes have been made to the manuscript, please provide justification.

We look forward to receiving your revised manuscript.

Kind regards,

Syed M. Faisal, Ph.D.

Academic Editor

PLOS ONE

Journal Requirements:

2. You indicated that ethical approval was not necessary for your study. We understand that the framework for ethical oversight requirements for studies of this type may differ depending on the setting and we would appreciate some further clarification regarding your research. Could you please provide further details on why your study is exempt from the need for approval and confirmation from your institutional review board or research ethics committee (e.g., in the form of a letter or email correspondence) that ethics review was not necessary for this study? Please include a copy of the correspondence as an ""Other"" file.

Additional Editor Comments:

When you submit the revised paper, please provide one "clean" copy and one copy where your changes are tracked (using the ‘track changes’ function in Word).

In your point-by-point responses to the reviewer, please state the page number and paragraph of the tracked manuscript where changes have been made as a result.

It can be helpful to tabulate your responses with columns labelled (left to right) as follows: Reviewer comments; author response and changes made; page number in revised, tracked paper where the change can be found.

IMPORTANT: Where a reviewer has asked for clarification, it is usually necessary to amend the manuscript as well as answering the question directly in the point-by-point response document - where no changes have been made to the manuscript, please provide justification.

Reviewers' comments:

Reviewer's Responses to Questions

**Comments to the Author**

1. Is the manuscript technically sound, and do the data support the conclusions?

Reviewer #1: Yes

Reviewer #2: Partly

2. Has the statistical analysis been performed appropriately and rigorously? 

Reviewer #1: Yes

Reviewer #2: Yes

3. Have the authors made all data underlying the findings in their manuscript fully available?

Reviewer #1: Yes

Reviewer #2: Yes

4. Is the manuscript presented in an intelligible fashion and written in standard English?

Reviewer #1: Yes

Reviewer #2: Yes

5. Review Comments to the Author

Reviewer #1: In this manuscript, Khang Le et al. explore the publicly available human cancer atlas to reveal the role of PRAME gene expression and its biological and clinicopathological significance in gliomas and established it as a potential prognostic biomarker for glioma.

The study has enormous translational importance, as they justified the prognostic signature in potential study cohorts for further clinical applications.

The authors should address the following issues to consider this manuscript, primarily standard. However, the description is sometimes concise, and the authors should provide more details.

Minor comments:

1.The general description of pertaining to glioma vs glioblastoma should be mentioned in

introduction section of the manuscript.

2. The author should discuss the role of IDH1-mutation in the prognosis of glioma.

3. Should make a diagram of signaling pathways that are upregulated and downregulated

in PRAME-positive gliomas and PRAME-negative gliomas.

Reviewer #2: In this manuscript, the authors investigated and claimed that PRAME positivity is an independent prognostic factor for poor outcomes. They suggested that its over-expression can indicate different biological and clinicopathological profiles, such as age, grade, and EGFR amplification, in adult glioma. High PRAME positivity was found to be solely associated with wtIDH glioma, glioma DNA methylation landscape, and poor outcome. While PRAME is an important molecular and clinicopathological marker for different cancers including adult glioma, the reviewer believes that the present work lacks novelty as similar conclusions have already been established in previous publications. Furthermore, the present work lacks a mechanistic approach, and several aspects of the manuscript still need to be properly addressed."

Major concerns:

1. It has already been established in previous publications that PRAME is expressed in adult glioma and is associated with poor prognosis. [Wei et al., J. Neurooncol. 2019 Apr; 142(2):375-384. doi: 10.1007/s11060-019-03110-5.] [Zhang et al., J. Neurooncol. 2008 May; 88(1): 65-76. doi: 10.1007/s11060-008-9534-4.] [Zhao et al., Ann Clin Lab Sci. 2022 Mar; 52(2): 185-195.] The present study primarily utilized public databases, and no immunohistochemical data were provided to support their claim. Please provide reasons to support the uniqueness and novelty of this article compared to previous publications.

2. According to the WHO 2021 guidelines, wtIDH gliomas are clustered together in grade 4, including diffuse astrocytoma, anaplastic astrocytoma, glioblastoma, and glioblastoma NOS. The present study claims a positive correlation between PRAME and wtIDH glioma, but it doesn't clearly explain the correlation between PRAME and these wtIDH gliomas. It is important to mention the expression status of PRAME in these specific groups of gliomas.

3. PRAME is highly immunogenic and serves as a robust target for immunotherapy. Certain drugs, such as Decitabine, can increase PRAME expression to enhance T-cell-mediated toxicity against GBM, which contradicts the findings of the present study. [Ma et al., Neuro-oncology. 2022 Dec; 24(12), 2093-2106]. The present study lacks an exploration of the immunological aspects of PRAME in gliomas. It is important to investigate the correlation between different immunological populations, pro-inflammatory signaling cascades, and PRAME.

4. Among 665 patients, 170 patients were PRAME-negative, and 495 patients were PRAME-positive. Moreover, within the PRAME-positive population, 481 had low PRAME expression, and only 14 had high PRAME expression. The study compared the PRAME-positive and -negative populations for various biological and clinicopathological parameters. However, the study didn't explain the biological and clinicopathological differences between the low and high PRAME-expressing populations and their relationship with patient survival. It would be worthwhile to compare the biological parameters between the 481 low and 14 high PRAME-expressing sample sets. Additionally, the number of samples in the high PRAME-expressing set is too small to draw conclusions. Increasing the sample size for the high PRAME-positive population is advisable.

5. In the Results section, the authors mentioned that PRAME-positive tumors expressed higher activity in the PI3K-AKT-mTOR, P53, and IL2-STAT5 pathways in gliomas. However, Fig 1C doesn't correlate with this statement, as it shows upregulation of KRAS, TGFb, and Notch signaling pathways in PRAME-positive tumors. Additionally, in Fig 3A, 3B, and 3C, it seems that the PRAME-positive population has longer survival than the PRAME-negative population, although statistical significance is lacking possibly due to the small sample size. Similarly, in Fig 4 and Supplementary Fig A & B, the PRAME low-expressing population tends to have longer survival than the PRAME-negative population. In Fig 3D, PRAME-positive mutant IDH1/2 shows higher survival than PRAME-negative wt-IDH1/2 population. Please provide a proper explanation of the data.

6. It has already been established that PRAME overexpression can inhibit the growth of breast cancer and leukemia. However, PRAME overexpression in glioma is associated with poor outcomes. The present study compared the hallmarks of apoptosis between PRAME-positive and -negative adult glioma but found no differences in apoptotic hallmarks between the two groups. Please explain these findings in detail with underlying mechanisms. Also, the specific role of PRAME in retinoic acid signaling in glioma malignancy is unknown. Since the present study lacks a mechanistic approach, it is important to consider the correlation of PRAME expression with retinoic acid signaling status in wtIDH gliomas.

7. Histone methylation or acetylation status, along with other epigenetic modifications, are important hallmarks of glioma development and progression. The present study did not establish any correlation between PRAME expression and histone methylation or acetylation status in glioma. Additionally, what is the correlation between G-CIMP and PRAME in glioma?

8. Radioresistance and chemoresistance are major obstacles in the successful treatment of adult glioma. Therefore, it would be advisable to investigate the correlation between PRAME expression and treatment resistance properties in adult gliomas.

Minor concerns:

1. Please add the references at the end of the third sentence in the Introduction section.

2. Please mention the figure/table numbers in the Results section at the end of the statements (a) "The association between PRAME expression and WHO grades was statistically significant" and (b) "Conversely, IDH1/2 mutations (87.9%) and ATRX mutations (39.8%) were significantly more common in PRAME-negative cases" under the sub-heading entitled "PRAME-positive gliomas had more advanced grades and adverse outcomes."

3. Properly mention the title of the X-axis in Fig 3A-D.

4. On the title page, it should be ORCID, not ORCHID. Please correct this typo error.

5. Please thoroughly check the manuscript for grammatical and typographical errors.

6. Please rephrase the second sentence of the Discussion section, as it appears confusing to readers."

6. PLOS authors have the option to publish the peer review history of their article (what does this mean?). If published, this will include your full peer review and any attached files.

Reviewer #1: **Yes: **Shadab Kazmi

Reviewer #2: No

---

## [Author Response · Author response to Decision Letter 0]

17 Jul 2023

To the editors: 

Thank you for processing our manuscript. In this revised version, we made modifications to the author board and analyses. Although there many alterations added, there are 3 main changes.

First, we regret to inform you that we removed our co-author, Kathryn Eschbacher, from the author board. It should be noted that there was no conflict of interest in this project and this removal was made solely based on the personal reasons of this author.

Secondly, we re-run the analyses by using our published preprocessing pipeline [PMID: 37253389] and additionally used data cbioportal for a higher quality of data. The results were similar but the data presentation can be different due to different random seeds of analysis. All the results related to Figures and Tables were revised accordingly.

Thirdly, we modified Table 2, focusing on 4 distinct multivariate analyses of PRAME status adjusted for clinical characteristics, genetic mutations (WGS-available data), TERT promoter mutations (WES-available data), and copy number alterations (7+/10-, EGFR amplification, and CDKN2A/B homozygous deletion). The reason for these modifications is due to the heterogeneity of data availability of different data types (clinical information, WGS, WES, and copy number array). Stratified survival analysis of PRAME status by new classifications was done independently by Kaplan-Meier analyses.

 

Reviewer #1: In this manuscript, Khang Le et al. explore the publicly available human cancer atlas to reveal the role of PRAME gene expression and its biological and clinicopathological significance in gliomas and established it as a potential prognostic biomarker for glioma. The study has enormous translational importance, as they justified the prognostic signature in potential study cohorts for further clinical applications.

Thank you for reviewing our manuscript.

The authors should address the following issues to consider this manuscript, primarily standard. However, the description is sometimes concise, and the authors should provide more details.

Minor comments:

1. The general description of pertaining to glioma vs glioblastoma should be mentioned in introduction section of the manuscript.

We added a few sentences describing the context of WHO classification regarding glioma and glioblastoma in Introduction as follows:

“Adult gliomas are a heterogeneous and common group of brain cancers with unclear cell-of-origin [5]. The biological profile of gliomas has been studied with respect to histology, epigenetic, genetic characteristics, cell-of-origin, and tumor microenvironment [6,7]. There are highly diverse oncogenic mechanisms contributing to gliomagenesis and tumor progression, including Wnt/𝛽-catenin [8], PI3K/Akt/mTOR [9], TGF-𝛽 [10], and mesenchymal transition, among many others [11]. Important genetic abnormalities affecting the prognosis of glioma patients consist of IDH1/2 mutations, CDKN2A/B homozygous deletion, EGFR amplification, TP53 mutations, ATRX mutations, TERT promoter mutations, and 7 gain 10 loss chromosomal abnormalities [12]. The recent World Health Organization (WHO) classification of Tumours of the Central Nervous System (CNS) emphasizes that glioma can be divided by IDH mutation and 1p/19q codeletion status. IDH-wildtype astrocytoma has more advanced clinicopathological progression and tumor with grade 4 is referred to as the “glioblastoma” category. Glioblastoma is diagnosed by the absence of IDH mutation and one of the high-grade features, including high-grade morphology, TERT promoter mutation, 7 gain/10 loss chromosomal abnormality, or EGFR amplification. However, the prognostic factors of glioma are still under investigation.”

2. The author should discuss the role of IDH1-mutation in the prognosis of glioma.

Thank you for your comments.

We added more details about IDH mutation in Introduction to clarify the background in the previous comment.

3. Should make a diagram of signaling pathways that are upregulated and downregulated

in PRAME-positive gliomas and PRAME-negative gliomas.

We added Figure 1D that describes pathways that are down-regulated or up-regulated regarding the PRAME expression status. 

 

Reviewer #2: In this manuscript, the authors investigated and claimed that PRAME positivity is an independent prognostic factor for poor outcomes. They suggested that its over-expression can indicate different biological and clinicopathological profiles, such as age, grade, and EGFR amplification, in adult glioma. High PRAME positivity was found to be solely associated with wtIDH glioma, glioma DNA methylation landscape, and poor outcome. While PRAME is an important molecular and clinicopathological marker for different cancers including adult glioma, the reviewer believes that the present work lacks novelty as similar conclusions have already been established in previous publications. Furthermore, the present work lacks a mechanistic approach, and several aspects of the manuscript still need to be properly addressed."

Thank you for reviewing our manuscript.

Major concerns:

1. It has already been established in previous publications that PRAME is expressed in adult glioma and is associated with poor prognosis. [Wei et al., J. Neurooncol. 2019 Apr; 142(2):375-384. doi: 10.1007/s11060-019-03110-5.] [Zhang et al., J. Neurooncol. 2008 May; 88(1): 65-76. doi: 10.1007/s11060-008-9534-4.] [Zhao et al., Ann Clin Lab Sci. 2022 Mar; 52(2): 185-195.] The present study primarily utilized public databases, and no immunohistochemical data were provided to support their claim. Please provide reasons to support the uniqueness and novelty of this article compared to previous publications.

Thank you for bringing these studies to our attention. We believe that the purpose of the aforementioned studies is different from our present work. We focused on the biological and clinicopathological investigation of PRAME expression in a more comprehensive way, including epigenetic, genetic, and gene expression. 

We added a paragraph to discuss this issue as follows:

“The significance of PRAME expression has been mentioned previously. Wu et al [20] developed a PRAME-containing formula for risk score, which was inferred from the regression model. This score quantified the risk of Karnofsky performance score, but not the prognosis itself. Therefore, the inference of PRAME prognostic value is plausible but weak. The other study by Zhang et al [21] mainly compared the PRAME expression between different types of brain tumors, including subtypes of astrocytic and non-astrocytic tumors. However, it can be difficult to conclude that PRAME expression is an independent prognostic factor by the current evidence. On the other hand, the goal of our study is to focus on the biological and clinicopathological characteristics of PRAME. Multiple analyses were performed to provide more concrete proof of the significance of this gene in glioma.”

2. According to the WHO 2021 guidelines, wtIDH gliomas are clustered together in grade 4, including diffuse astrocytoma, anaplastic astrocytoma, glioblastoma, and glioblastoma NOS. 

We believe that Glioblastoma, IDH-wildtype is defined when there is one or more of the following features: microvascular proliferation, necrosis, TERT promoter mutation, EGFR gene amplification, and 7+/10- chromosome copy-number changes according to WHO classification 5th edition. Therefore, astrocytic tumors with IDH wildtype status and no other feature are ill-defined and should be classified as Astrocytoma, NOS. Therefore, we revised the classification in a more detailed way, following our recently published paper [PMID: 37253389].

We added details about our revised data processing pipeline as follows:

“Data processing

 The Human Cancer Atlas (TCGA) database consists of many datasets. We extracted cases from the TCGA-GBM and TCGA-LGG projects. Only cases with available gene expression profiles (GEP) and primary tumors (no recurrent or metastatic tumors) were included in the study. To adapt to the new WHO classification, cases with grades 2/3 in the previous studies [13,14] were reclassified into grade 4 as follows: (1) the presence of both IDH1/2 mutation and CDKN2A/B homozygous deletion, or (2) the absence of IDH1/2 mutations and the presence of at least one of the following abnormalities: TERT promoter mutation, EGFR amplification, and 7 gain 10 loss chromosomal abnormality. Other histopathological grading features such as microvascular proliferation and pseudopalisading necrosis were assumed to be included in previously evaluated grade IV gliomas of the original studies [13,14]. Tumors with the absence of IDH1/2 mutations and no other high-grade morphological and genetic features mentioned above were reclassified as Astrocytoma, Not Otherwise Specified (NOS). Tumors with the presence of both IDH1/2 mutations and 1p/19q codeletion were categorized as Oligodendroglioma. Mixed glioma was re-distributed into new categories based on IDH1/2 mutation and 1p/19q codeletion status. This reclassification was published in our previous paper [15]. The difference in data processing between this study and our previous one was that we used cbioportal for cancer genomics (https://www.cbioportal.org) datasets that are related to TCGA-GBM and TCGA-LGG projects, including (1) Brain Lower Grade Glioma (TCGA, Firehose Legacy), (2) Glioblastoma Multiforme (TCGA, Firehose Legacy), and (3) Merged Cohort of LGG and GBM (TCGA, Cell 2016). This difference led to a slight inconsistency in the total number of glioma patients.”

The present study claims a positive correlation between PRAME and wtIDH glioma, but it doesn't clearly explain the correlation between PRAME and these wtIDH gliomas. It is important to mention the expression status of PRAME in these specific groups of gliomas.

We added a boxplot (Figure 1E) to illustrate the PRAME expression status in each revised category of glioma.

3. PRAME is highly immunogenic and serves as a robust target for immunotherapy. Certain drugs, such as Decitabine, can increase PRAME expression to enhance T-cell-mediated toxicity against GBM, which contradicts the findings of the present study. [Ma et al., Neuro-oncology. 2022 Dec; 24(12), 2093-2106]. The present study lacks an exploration of the immunological aspects of PRAME in gliomas. It is important to investigate the correlation between different immunological populations, pro-inflammatory signaling cascades, and PRAME.

Thank you for your information.

We added a section of analysis named “Analysis of the association between tumor microenvironment (TME) and PRAME expression” as follows:

“Analysis of the association between tumor microenvironment (TME) and PRAME expression

Given that PRAME is associated with cytotoxic T-cell activation and killing in glioblastoma [15], we compared the immunologic cell population between PRAME-negative and PRAME¬-positive gliomas. GSVA of cell type-specific gene sets [16] was performed to calculate the activity (ES) of 17 cell types, including B cells, T cells, T helper, Th1, Th2, TFH, Th17, Treg, CD8 T cells, T gamma delta, cytotoxic cells, NK cells, dendritic cells, eosinophils, macrophages, mast cells, and neutrophils (Figure 5). There were increased activities of T cells (p<0.001), Th2 (p<0.001), Th17 (p<0.001), cytotoxic cells (p=0.028), macrophages (p<0.001), and neutrophils (p<0.001) in PRAME-positive gliomas while there was reduced activities of TFH (p<0.001) and CD8 T cell (p<0.001).”

We also added a corresponding discussion paragraph:

“PRAME can also be associated with glioma TME. A previous study showed that Decitabine can increase PRAME expression and, thus, enhance the T-cell-mediated cytotoxicity, which makes PRAME an interesting target for immunotherapy [15]. However, TME of cell line can be difficult to interpret because the stromal or microenvironment context of cancer in vivo is different from that of cell line condition. Our study showed that PRAME higher expression was related to increased cytotoxic cell, macrophage, and neutrophil activity but it was also associated with many immune modulating cells such as Th2, Th17, and TFH. Therefore, PRAME expression is in a complicated relationship with many immunologic cell populations, not only cytotoxic T cells.”

4. Among 665 patients, 170 patients were PRAME-negative, and 495 patients were PRAME-positive. Moreover, within the PRAME-positive population, 481 had low PRAME expression, and only 14 had high PRAME expression. The study compared the PRAME-positive and -negative populations for various biological and clinicopathological parameters. However, the study didn't explain the biological and clinicopathological differences between the low and high PRAME-expressing populations and their relationship with patient survival. It would be worthwhile to compare the biological parameters between the 481 low and 14 high PRAME-expressing sample sets. Additionally, the number of samples in the high PRAME-expressing set is too small to draw conclusions. Increasing the sample size for the high PRAME-positive population is advisable.

Thank you for your suggestion. However, it is quite difficult to define a cutoff of PRAME overexpression. This is emperically determined by the data at hand and defined by >1, >3, or >5 FPKM. For example, a previous study used FPKM>1 to define higher expression [PMID: 27577089]. Therefore, we believe that it is better to follow these recommendations. In our data, a vast number of cases had PRAME read count<1. Therefore, >1 FPKM is an appropriate cutoff for determining high expression or overexpression.

As the reviewer mentioned, the sample size of PRAME-overexpressing cohort was small to create reliable results. Therefore, to avoid multiple hypothesis problem, we only performed survival analysis for this cohort. We additionally applied Benjamini-Hochberg correction for the pairwise comparison in this survival analysis to solidify the result. The analysis of other biological parameters, which requires a lot of hypothesis testing, can be performed but it can lead to false positive results and misleading conclusions. Therefore, we believe that it is better not to over-do the investigation of the PRAME-overexpressing cohort.

5. In the Results section, the authors mentioned that PRAME-positive tumors expressed higher activity in the PI3K-AKT-mTOR, P53, and IL2-STAT5 pathways in gliomas. However, Fig 1C doesn't correlate with this statement, as it shows upregulation of KRAS, TGFb, and Notch signaling pathways in PRAME-positive tumors. Additionally, in Fig 3A, 3B, and 3C, it seems that the PRAME-positive population has longer survival than the PRAME-negative population, although statistical significance is lacking possibly due to the small sample size. Similarly, in Fig 4 and Supplementary Fig A & B, the PRAME low-expressing population tends to have longer survival than the PRAME-negative population. In Fig 3D, PRAME-positive mutant IDH1/2 shows higher survival than PRAME-negative wt-IDH1/2 population. Please provide a proper explanation of the data.

Thank you for pointing out this observation. In Figure 3A, PRAME-positive and PRAME-negative glioma patients had difference in survival patterns (p<0.001). We concur with you that the PRAME-negative cohort survived better in the >10 years of follow-up. Unfortunately, we do not have a clear explanation for this observation. Although it seems that PRAME-negative cohort deceased sooner than PRAME-positive cohort in the period of >10 years of follow-up, there may be many confounding factors because of the long follow-up time. Another confounding factor is sample size. The size of PRAME-positive cohort was much larger than PRAME-negative one. Therefore, the positive cohort was likely to have more outliers than the negative cohort. Generally, first years of follow-up are more reliable prognostic indicator compared to longer term of follow-up. To avoid this confusing phenomenon, we limited the x-axis from 0 to 10 years of follow-up, which eliminates such observation of the outliers.

6. It has already been established that PRAME overexpression can inhibit the growth of breast cancer and leukemia. However, PRAME overexpression in glioma is associated with poor outcomes. The present study compared the hallmarks of apoptosis between PRAME-positive and -negative adult glioma but found no differences in apoptotic hallmarks between the two groups. Please explain these findings in detail with underlying mechanisms. Also, the specific role of PRAME in retinoic acid signaling in glioma malignancy is unknown. Since the present study lacks a mechanistic approach, it is important to consider the correlation of PRAME expression with retinoic acid signaling status in wtIDH gliomas.

Thank you for your suggestion. We experimentally run pathway analysis to investigate the activity of retinoic acid signaling (the full gene set of this pathway is in the link below). The enrichment score (ES) of PRAME-negative and PRAME-positive glioma were compared by two-sample independent t-test, which was similar to other pathways. There was no significant difference between PRAME-negative and PRAME-positive samples (p=0.446). Therefore, the role of PRAME in glioma may not be related to its suppression of retinoic acid signaling. However, we believe that it is better not to include this analysis into the manuscript because this analysis may not be relevant to the main stream of analysis and it can also cause confusion to readers.

The link to retinoic pathway: 

https://www.gsea-msigdb.org/gsea/msigdb/human/geneset/REACTOME_SIGNALING_BY_RETINOIC_ACID.html

7. Histone methylation or acetylation status, along with other epigenetic modifications, are important hallmarks of glioma development and progression. The present study did not establish any correlation between PRAME expression and histone methylation or acetylation status in glioma. Additionally, what is the correlation between G-CIMP and PRAME in glioma?

Thank you. The lack of histone modification data is the limitation of the present study. We added a few sentences to the Discussion as follows:

“However, there were limitations in the present study. First, selection bias was a potential problem because this study used a public database. Second, our findings of PRAME positivity in IDH-mutant glioma were not significant potentially due to small samples, and, thus, sampling error. Therefore, a larger study of PRAME expression on IDH-mutant glioma can be helpful to examine the biological and clinicopathological relevance of PRAME positivity in these brain tumors. Third, protein expression data was not fully available and, therefore, cannot be analyzed. PRAME gene expression may be different from PRAME protein expression, which can be practically evaluated by immunohistochemistry. Hence, immunohistochemical studies are required to validate PRAME prognostic significance at the protein level. Additionally, data about histone modification is not available in TCGA-LGG and TCGA-GBM projects. Therefore, we were not able to analyze the relationship between PRAME expression and this epigenetic regulation. Finally, to our knowledge, there was no available information of chemotherapy and radiotherapy resistance in TCGA datasets. Hence, we were not able to investigate the relationship between PRAME expression and treatment response.”

In addition, we created Figure 2E to demonstrate the relationship between G-CIMP and PRAME expression in glioma.

To avoid confusion, we eliminate the analysis of t-SNE methylation clusters. We instead used the published methylation CIMP clusters and performed chi-square test. We modified the methylation Results section as follows:

“In the study cohort, there were a total of 476 tumors with available data about 450K DNA methylation. We performed t-SNE dimension reduction to explore the differences in the distribution of PRAME-negative and PRAME-positive glioma in DNA methylation hyperspace. This DNA methylation space can be interpreted as the reduced representation of the CpG methylation landscape. We also included the Glioma CpG Island Methylator Phenotype (G-CIMP), which was published in a previous paper [33941250]. In this study, 476 tumors in DNA methylation space can be relatively divided into two unsupervised tSNE clusters, small (right, lower corner) and large clusters (left and upper part) (Figure 2A-D). The small cluster densely consisted of PRAME-low and PRAME¬-overexpressing samples while the larger cluster had a significant portion of PRAME-negative samples (Figure 2A). The revised subtype (Figure 2B), CIMP clusters (Figure 2C), and IDH status (Figure 2D) were strongly associated with these 2 clusters. Figure 2E shows a heatmap of distribution of PRAME expression status within CIMP clusters. There were significant difference in the distribution of PRAME expression status (chi-square test, p<0.001). This discrimination can be seen in LGm6-GBM (6/12 vs. 7/484 vs. 0/170), classic-like (1/12 vs. 63/484 vs. 4/170), and mesenchymal-like (2/12 vs. 87/484 vs. 9/170).”

8. Radioresistance and chemoresistance are major obstacles in the successful treatment of adult glioma. Therefore, it would be advisable to investigate the correlation between PRAME expression and treatment resistance properties in adult gliomas.

Thank you for your comment. However, there was no information about chemotherapy and radiotherapy resistance in the TCGA datasets. Therefore, we listed your comments as a limitation of our study.

Minor concerns:

1. Please add the references at the end of the third sentence in the Introduction section.

We added citations at the end of the third sentence in Introduction section.

2. Please mention the figure/table numbers in the Results section at the end of the statements (a) "The association between PRAME expression and WHO grades was statistically significant" and (b) "Conversely, IDH1/2 mutations (87.9%) and ATRX mutations (39.8%) were significantly more common in PRAME-negative cases" under the sub-heading entitled "PRAME-positive gliomas had more advanced grades and adverse outcomes."

Thank you. The referred paragraph was written as a description of Table 1 and it is mentioned in the first sentence of this paragraph:

“Table 1 summarizes the clinicopathological characteristics of PRAME-negative and PRAME-positive gliomas.”

3. Properly mention the title of the X-axis in Fig 3A-D.

We changed the title of the x-axis from “Time” to “Follow-up Time (Years)”.

4. On the title page, it should be ORCID, not ORCHID. Please correct this typo error.

We corrected this error.

5. Please thoroughly check the manuscript for grammatical and typographical errors.

Thank you for your comments.

6. Please rephrase the second sentence of the Discussion section, as it appears confusing to readers.

We modified the sentence “In gene expression analysis, we explored PRAME read counts and found that PRAME expression was generally low in these brain tumors. A significant number of tumors showed no PRAME expression.” into “In gene expression analysis, there was a large number of gliomas showing no PRAME expression while a few numbers of tumors possessed high levels of PRAME expression. The remaining tumors generally showed low PRAME expression.”

---

## [Editor Report · Decision Letter 1]

11 Aug 2023

Molecular and clinicopathological implications of PRAME expression in adult glioma

PONE-D-23-03723R1

Dear Dr. Kondo,

We’re pleased to inform you that your manuscript has been judged scientifically suitable for publication and will be formally accepted for publication once it meets all outstanding technical requirements.

Kind regards,

Syed M. Faisal, Ph.D.

Academic Editor

PLOS ONE
---

## [Editor Report · Acceptance letter]

25 Sep 2023

PONE-D-23-03723R1 

Molecular and clinicopathological implications of *PRAME* expression in adult glioma 

Dear Dr. Kondo:

I'm pleased to inform you that your manuscript has been deemed suitable for publication in PLOS ONE. Congratulations! Your manuscript is now with our production department. 

Kind regards, 

on behalf of

Dr. Syed M. Faisal 

Academic Editor

PLOS ONE